

# Code stylometry vs formatting and minification

Stefano Balla[1], Maurizio Gabbrielli[1] and Stefano Zacchiroli[2]

[1] DISI, University of Bologna, Bologna, Italy
[2] LTCI, Télécom Paris, Institut Polytechnique de Paris, Palaiseau, France

## ABSTRACT

The automatic identification of code authors based on their programming styles—known as authorship attribution or code stylometry—has become possible in recent years thanks to improvements in machine learning-based techniques for author recognition. Once feasible at scale, code stylometry can be used for well-intended or malevolent activities, including: identifying the most expert coworker on a piece of code (if authorship information goes missing); fingerprinting open source developers to pitch them unsolicited job offers; de-anonymizing developers of illegal software to pursue them. Depending on their respective goals, stakeholders have an interest in making code stylometry either more or less effective. To inform these decisions we investigate how the accuracy of code stylometry is impacted by two common software development activities: code formatting and code minification. We perform code stylometry on Python code from the Google Code Jam dataset (59 authors) using a code2vec-based author classifier on concrete syntax tree (CST) representations of input source files. We conduct the experiment using both CSTs and ASTs (abstract syntax trees). We compare the respective classification accuracies on: (1) the original dataset, (2) the dataset formatted with Black, and (3) the dataset minified with Python Minifier. Our results show that: (1) CST-based stylometry performs better than AST-based (51.00%→68%), (2) code formatting makes a significant dent (15%) in code stylometry accuracy (68%→53%), with minification subtracting a further 3% (68%→50%). While the accuracy reduction is significant for both code formatting and minification, neither is enough to make developers non-recognizable via code stylometry.

# INTRODUCTION

It has become possible in recent years to automatically identify with high accuracy the author of a given piece of software source code. Doing so is referred to as, interchangeably in the literature (*e.g.*, *Islam et al., 2015*; *Wang, Ji & Wang, 2018*), *authorship attribution* or *code stylometry*—we use the two terms interchangeably in the following. Several techniques for code stylometry exist, from pioneering ones (*Oman & Cook, 1989*; *Islam et al., 2015*; *Tereszkowski-Kaminski et al., 2022*) based on explicit formal rules that measure stylistic features of source code writing, to more recent and capable approaches based on machine learning (*Alsulami et al., 2017*; *Kurtukova, Romanov & Shelupanov, 2020*) and

Corresponding author
Stefano Balla, stefano.balla2@unibo.it

embeddings (*Alon et al., 2019*; *Bogomolov et al., 2021*; *Azcona et al., 2019*; *Kovalenko et al., 2020*). (See 'Related Work' for an overview of the state-of-the-art of code stylometry techniques.)

Code stylometry has important applications in software engineering, ranging from code clone detection (*Büch & Andrzejak, 2019*; *Ye et al., 2020*) and retrieving authorship information that went missing (*Ou et al., 2023*) to productivity enhancements (*Kovalenko et al., 2020*; *Azcona et al., 2019*). The ability to automatically identify source code authors, however, also comes with privacy concerns when authors *did not want* to be identified in the first place (*Simko, Zettlemoyer & Kohno, 2018*; *Ucci, Aniello & Baldoni, 2017*; *Rocha et al., 2017*; *Yang et al., 2022*)—which might be so for arguably good reasons (*e.g.*, hiding personal involvement in the development of a censorship avoidance open source product when living in a totalitarian state) or bad reasons (*e.g.*, illegal or unethical activities such as plagiarism detection evasion). This creates a typical "arms race" situation in which evaders look for how to be *less* recognizable (while still distribute source code, whose availability is the premise of code stylometry), while detectors look for how to improve detection performances.

The state of knowledge on the impact of various kinds of source code manipulation on authorship attribution is, however, fairly limited (*Kurtukova, Romanov & Shelupanov, 2020*; *Li et al., 2022*). Adversarial machine learning scenarios have been recently considered in *Li et al. (2022)*, but the impact of source-to-source code transformations that are routinely applied by software developers—such as code formatting and code minification—on the accuracy of code stylometry remains largely unexplored. In this work we contribute to fill this gap by benchmarking code stylometry against code formatting and code minification, in a controlled experiment on a uniform dataset and with a common experimental methodology. The first practice, *code formatting*, consists in making source code adhere to a given coding style guide and is often implemented by delegation to an automated tool, possibly integrated with an IDE (Integrated Development Environment). The second practice, *code minification*, consists in automatically altering semantic-meaningless parts of the code such as blanks and variable names, with the goal of reducing code size to a bare minimum, *e.g.*, to minimize its distribution time to a web browser.

Both practices, code formatting and minification, are source-to-source transformations that do not alter code semantics, but modify most or all (depending on the practice) of the surface aspects of code—lexical, layout, syntactic—that fall within common definitions of *programming style* (*Oman & Cook, 1989*; *Islam et al., 2015*). Note how most (or all) of these changes do not affect the abstract syntax tree (AST) of code. For instance, it is possible to thoroughly re-indent a piece of code, changing its spacing, obtaining the same after-parsing AST. The corresponding *concrete* syntax tree (CST), however, would be significantly altered by the same source-to-source transformation. Please refer to Fig. 1 for a visual distinction between an AST and a CST.

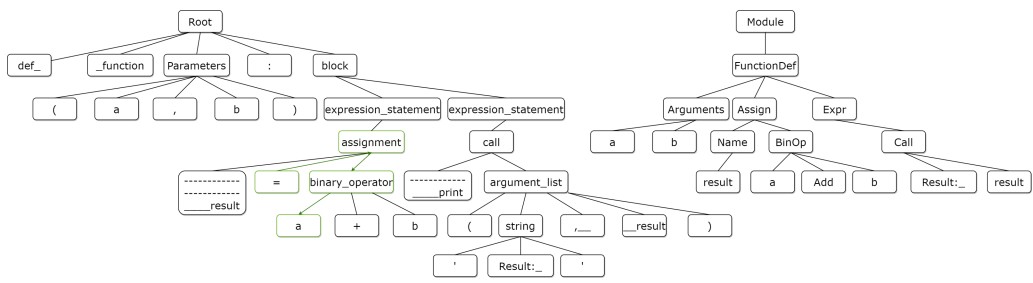

**Figure 1** **Concrete Syntax Tree (CST) representation (on the left) and corresponding Abstract Syntax Tree (AST, on the right) of the Python code snippet in Listing 1.** Highlighted in bold and green on the CST is the CST path: $[=, \uparrow, \text{assignment}, \downarrow, \text{binary\_operator}, \downarrow, a]$, of length 3, joining the " =" and "a" terminals.

*Contributions.* The first contribution of this work is an answer to the following research question, currently unanswered in the code stylometry literature:

**RQ 1 (AST *vs* CST)** *What is the impact of using CSTs (instead of ASTs) on the ability to automatically identify authors of Python source code?*

Answering this question provides quantitative insights on what makes author classifiers (in)effective. Depending on the findings, future code stylometry work should concentrate on AST- *vs* CST-based code representations.

For the next two contributions of this article we move to the impact of code formatting and minification on code stylometry, answering the following research questions:

**RQ 2 (code formatting)** *What is the impact of code formatting on the ability to automatically identify authors of Python source code?*

**RQ 3 (code minification)** *What is the impact of code minification on the ability to automatically identify authors of Python source code?*

We address all the stated research questions by performing code stylometry using an author classifier based on code2vec (*Alon et al., 2019*) to classify Python source code authored by 59 participants from the Google Code Jam dataset (*Google, 2023*), one of the most popular datasets in the literature for this task (*Islam et al., 2015*; *Tereszkowski-Kaminski et al., 2022*; *Alsulami et al., 2017*; *Kurtukova, Romanov & Shelupanov, 2020*; *Bogomolov et al., 2021*). We compare the classification accuracies obtained on (1) the original unmodified dataset using both CSTs and ASTs, (2) the dataset formatted with Black (*Langa, 2024*) using CSTs only, and (3) the dataset minified with Python Minifier (*Flook, 2024*) using CSTs only. The two chosen tools, Black and Python Minifier, are popular and state-of-the-art tools for the respective tasks (*Hart et al., 2023*; *Flook, 2024*).

*Key findings.* Experimental results show that:

- Moving from ASTs to CSTs leads to an improvement in code stylometry accuracy from 51%to 68%, suggesting that to maximize author recognizability CST-based approaches are preferable.
- Code formatting negatively impacts code stylometry accuracy (68% →53%).

- Code minification negatively impacts code stylometry accuracy, further subtracting 3% (68% →50%). While the reduction is significant for both techniques, the developers remain recognizable *via* code stylometry.
- While there are accuracy benefits in including concrete syntax features when performing code stylometry, those benefits are neutralized in code that is subject to formatting and/or minification.

*Article structure.* 'Background' provides the necessary background to understand the technical details of the study; it might be skipped by readers familiar with code stylometry. 'Related Work' discusses related work and compares it with the contributions of this article. The experimental methodology for this study is detailed in 'Methodology'. Results are presented in 'Results' and discussed in 'Discussion', also covering threats to the validity of our experiments. Conclusion and future work are discussed in 'Conclusion'.

*Data availability.* A complete replication package for the experiments described in this article is now available on Zenodo (https://doi.org/10.5281/zenodo.10528246).

## RELATED WORK

*Oman & Cook (1989)* is one of the pioneering works on code stylometry. It defined a set of formal rules for capturing stylistic features of source code and performed automated authorship attribution using cluster analysis. This approach was expanded upon by *Islam et al. (2015)*, who introduced the definition of code stylometry as we understand it today, separating lexical, layout, and syntactic features (see 'Background'). Their methodology incorporated the use of AST, advancing beyond the reliance of language-specific features. Using a tree classifier, their approach yielded a 98% accuracy rate in recognizing 250 authors for full-length source code files written in C, and 53.91% for 229 authors of Python files.

*Dauber et al. (2019)* observed that the efficacy of this method diminishes drastically as the number of lines of code (SLOCs) decreases. They applied the technique of *Islam et al. (2015)* to C snippets fragments with an average length of 4.9 SLOCs (*versus* 70 SLOCs in Caliskan et al.'s study), obtaining an accuracy of 48.8%. *Tereszkowski-Kaminski et al. (2022)* utilized a tree-based technique on ASTs of C++ source code obtained from the Google Code Jam (GCJ) dataset and real-world open source projects from GitHub. On 100 authors from the GCJ dataset they obtained 95% accuracy; on the same amount of authors of real-world GitHub accuracy was 60%.

In comparison with the studies described thus far our approach: (1) uses the Python part of the GCJ dataset; (2) is based on CSTs (Concrete Syntax Trees) rather than ASTs, which is required to appreciate changes induced by code formatting and minification; (3) is independent from specific details of a given programming language, which are neither used as features for recognition nor a strict implementation-level dependency (we rely on tree-sitter, which is programming language agnostic); (4) uses machine learning (ML) techniques for classification, as introduced by more recent works.

*Alsulami et al. (2017)* introduced the use of ML for authorship attribution. Utilized a Bi-LSTM method on Python ASTs from the GCJ dataset, authors obtained an accuracy of 88.86% for complete Python files by 70 distinct authors. We differ from *Alsulami et al. (2017)* in the fact that we also consider layout features (according to Caliskan classification), which their approach ignores. These features are crucial by Caliskan's definition for quantifying the impact of code formatting and minification. *Yang et al. (2017)* obtained a code stylometry accuracy of 91.1% on a Java dataset of 40 authors, leveraging all of lexical, layout, and syntactic features. Departing from previous ML-based work they utilized particle swarm optimization for training their ML classifier. These studies emphasize the effectiveness of neural networks for authorship recognition, surpassing that of tree-based methods. *Li et al. (2022)* considered the adversarial scenario in which authors would like to evade recognition based on neural network techniques. In response, they introduced RoPGen, a framework that enhances the robustness of ML-based code stylometry through data augmentation during the training phase. Our study complements theirs, providing the first quantification of how much commonly used software engineering techniques (code formatting, minification) contribute to stylometry evasion.

*Wang, Ji & Wang (2018)* offer a distinct perspective on code style. Unlike the conventional static understanding of it, shared by our work, which focuses solely on source code features, they introduce the notion of dynamic style, derived from runtime execution data. *Gull, Zia & Ilyas (2017)* further expand upon the conventional understanding of code style, by incorporating code smells in their analysis.

A prominent subset of ML-based techniques for code stylometry employs the code2vec architecture (*Alon et al., 2019*) to represent source code as vectors. *Bogomolov et al. (2021)* utilized this approach for authorship attribution of complete source code files from the GCJ dataset, achieving 97.9% accuracy for Java and 72.3% for Python. code2vec has also been used to spatially represent authors based on their programming style (*Azcona et al., 2019*; *Kovalenko et al., 2020*). Our work follows in the step of these works, by using a code2vec-based classifier as baseline. Where we diverge is in the use of CSTs, rather than ASTs, as input before code2vec embedding.

*Kurtukova, Romanov & Shelupanov (2020)* investigated the impact of specific development practices on code stylometry using a hybrid neural network. They considered code obfuscation of Python code using Opy and PyArmor as tools, as well as mandatory adherence to strict coding guidelines in a project, with the notable example of the Linux kernel. Their results show that authors from projects that adopt code obfuscation and/or strict coding guidelines are more difficult to recognize than authors in projects that use neither. Our results belong to the same problem space—quantifying the effect of specific practices on author recognizability—with two main differences. First, our methodology allows to decorrelate the effect of the practices from the context: it is the *same code by the same authors*, before and after formatting and minification, that undergoes code stylometry. Second, we explore different development practices: minification (less invasive and more commonly used than obfuscation) and code formatting.

## BACKGROUND

In this section we briefly review some background notions upon which the rest of the article builds.

### Stylometry

*Stylometry* consists in the computational and statistical examination of writing style, based on the assumption that authors consistently showcase identifiable and distinct patterns in their writing (*The Classical Review, 1897*). Note that stylometry is not specific to source code or software. *Code stylometry* is a specialization of stylometry to the writing of software source code. It employs a range of techniques to discern programming styles, based on different metrics. Our choice of metrics and techniques for code stylometry follows in the steps of the foundational work by *Islam et al. (2015)*, who considered three classes of features that encapsulate programming style:

- **Lexical features** related to how small parts of source code, like words and characters, are used.
- **Syntactic features** related to how a developer organizes grammatical structures of the language, *i.e.,* the shape and structure of abstract syntax trees (ASTs) obtained after code parsing.
- **Layout features** related to the "graphic" layout of source code, as the result of how spacing, indentation, and block/line lengths are used.

### Code representation

In machine learning based code stylometry, many approaches, including (*Alon et al., 2018*; *Chen & Monperrus, 2019*; *Wei & Li, 2017*; *White et al., 2016*), and those discussed in 'Related Work', employ ASTs for code representation. Such a choice preserves all syntax features (by definition) and, to a lesser extent, lexical features of the original source code before parsing, but does not capture layout features. Our research questions relate to how layout and lexical features impact the recognizability of authors *via* code stylometry techniques. As such, we cannot use code representations like ASTs that abstract over them.

Hence, deviating from the majority of existing literature on code stylometry, we use *CSTs (concrete syntax trees)* as code representations. Both code formatting and minification change concrete syntax trees in measurable ways, leading to different inputs fed to stylometry-based author classifiers.

CST are tree representations of a context-free grammar (*Wile, 1997*). They are formal representations that show how the compiler understands the code.

**Definition 1 (Concrete Syntax Tree):**  *Given a grammar, $G = (NT, T, R, S)$, a concrete syntax tree (or parse tree) is an ordered tree in which:*

- *Each node is labeled with a symbol in $NT \cup T \cup \{\epsilon\}$;*
- *The root is labeled with $S$;*
- *Each non-leaf node is labeled with a symbol in $NT$;*

Listing 1: Python code snippet (badly formatted).

```
␣␣def␣function(a,b):                                              1
                                                                 2
                                                                 3
␣␣␣␣result=a+b                                                    4
                                                                 5
                                                                 6
␣␣␣␣print('Result:␣',␣␣result)                                    7
```

- If a node has label $A \in NT$ and its children are $m_1, \ldots, m_k$ labeled respectively with $X_1, \ldots, X_k$, where $X_i \in NT \cup T$ for all $i \in [1,k]$, then $A \longrightarrow X_1, \ldots, X_k$ is a production of R;
- If a node has a label $\epsilon$, then that node is unique child of its parent $A$, and $A \longrightarrow \sigma$ is a production in R.

Given X, the set of all possible concrete syntax tokens in the source code language, $\phi : T \longrightarrow X$ is a function mapping terminal nodes to concrete syntax tokens.

Intuitively a CST renders code as a hierarchical structure, where each node relates to distinct code constructs—like class definitions, function definitions, or variable assignments. These nodes precisely capture all facets of the code, ranging from overarching syntactic elements to finer-grained stylistic, lexical, and layout specifics. This granularity ensures that each node retains comprehensive syntactic and stylistic information from the original code.

```
=,68efd4a5067b71cd72958a574638a3920307c0a623c3f3ff478938b4ba65d21d,a
(,ee1e5ece205f4f74f66abd2bc93c4554d6c884c32ac0dd9b7a7665281d635a23,Result:_
a,20be399d2eb74dae602b9c0b09683c2ca6a6e71c1f25c95e77a4e8ed4e39df49,b
```

As an example, consider the (badly formatted, on purpose) Python code snippet shown in Listing 1. The corresponding CST is shown in Fig. 1.

**Definition 2 (CST path):** A CST path of length k is a sequence $n_1, d_1, \ldots, n_k, d_k, n_{k+1}$, where $n_1$ and $n_{k+1} \in T$ are terminals, whereas for $i \in [2..k] : n_i \in NT$ are non-terminals, and $\forall i \in [1..k] : d_i \in \{\uparrow, \downarrow\}$ are movement directions (either up or down in the tree).

If $d_i = \uparrow$, then: $n_i$ is a child of $n_{i+1}$; whereas if $d_i = \downarrow$, then: $n_i$ is the parent of $n_{i+1}$. For a CST path p, we use start(p) to denote $n_1$ the starting terminal of p, and end(p) to denote $n_{k+1}$ its final terminal.

As an example look back at Fig. 1, where a CST path of length 3, joining the "=" and "a" terminals, is highlighted in bold and green.

**Definition 3 (path context):** Given a CST path p, its path context is a triplet $\langle x_s, p, x_t \rangle$ where $x_s = \phi(start(p))$ and $x_t = \phi(end(p))$ are the values associated with the start and end terminals of p.

That is, a path context describes two actual tokens from input source code, together with the (CST) path joining them.

**Definition 4 (bag of path contexts):** Given an input source code snippet, its bag of path contexts is the set of all its path contexts.

```
=,68efd4a5067b71cd72958a574638a3920307c0a623c3f3ff478938b4ba65d21d,a
(,ee1e5ece205f4f74f66abd2bc93c4554d6c884c32ac0dd9b7a7665281d635a23,Result:_
a,20be399d2eb74dae602b9c0b09683c2ca6a6e71c1f25c95e77a4e8ed4e39df49,b
```

**Figure 2** **Subset of the bag of path contexts for the code snippet in Listing 1.** Each line corresponds to a path context in the format $\langle x_s, \text{sha256}(p), x_t \rangle$. The first path in the example corresponds to the path highlighted in Fig. 1.

Intuitively, the bag of path contexts of a source code snippet is a representation of it that adeptly retains the lexical, layout, and syntactic features inherent in the code.

For practical purposes, and following code2vec (_Alon et al., 2019_), the path component of a path context (second element of the triple) is first rendered as a sequence of non-terminal nodes and then hashed using the SHA-256 algorithm to obtain a unique identifier for the path, effectively compressing it while retaining its uniqueness. This results in a representation of path contexts in the format: $\langle x_s, sha256(p), x_t \rangle$.

As an example, Fig. 2 shows a subset of the bag of path contexts of the code snippet in Listing 1.

## Code formatting

Adherence to a common coding convention (_Smit et al., 2011_) is a best practice in software development, particularly in collaborative software development. Code formatting is the subset of a coding convention that refers to the way the code is styled and organized. It includes factors such as indentation, use of spaces _vs_ tabs, placement of brackets, line length, _etc_. These factors can have a profound impact on the readability, maintenance, and even the functionality of the code (_Oliveira et al., 2023_).

Consistent code formatting improves code legibility and avoids distractions induced by style inconsistencies when reading code. _Code formatting_ is a way to ensure consistent code formatting where formatting decisions are not taken interactively by developers, but instead automatically enforced by tooling, often integrated into IDEs.

To provide an experimental answer to RQ 2 (code formatting) we consider the case of the Python programming language using Black (_Langa, 2024_) as fully automated code formatter. Black is a very popular code formatter for Python, which favors consistent styling over configurability—for those reasons it is often referred to as an "opinionated" and/or "uncompromising" code formatter.

Among the changes that Black can apply to input code to make it fit the target style there are: indentation, spacing (addition/suppression of blank lines), string quotes, literals, parentheses, _etc_. For a comprehensive description of the modifications Black can apply we refer the reader to the tool documentation at https://black.readthedocs.io/. Nonetheless, Black guarantees the preservation of code execution semantics across code reformatting.

As an example, Listing 2 displays the result of formatting the code snippet of Listing 1 with Black.

Listing 2: Python code snippet from Listing 1, after formatting with Black. Note how horizontal and vertical blanks have been altered, as well as string quotes.

```
  def function(a, b):               1
    result = a + b                  2
                                    3
    print("Result: ", result)      4
```

Listing 3: Python code snippet from Listing 1, after minification with Python Minifier. Note how whitespace has been removed and the longer identifiers "function" and "result" have been renamed to "A".

```
  def A(a,b):A=a+b;print('Result: ',A)     1
```

## Code minification

*Code minification* is an optimization technique used to reduce the size of software that must be distributed in source code form. It is common place in Web development, where minimizing the amount of source code that a browser needs to download can significantly speed up page rendering time. Minification entails the removal of semantically meaningless elements from source code, such as white spaces, line breaks, and comments, as well as more invasive changes such as renaming identifiers (*e.g.*, variable and class names) to more concise alternatives.

As per code formatting, minification must be neutral with respect to code semantics in order to be applicable. Contrary to code formatting, code minification *reduces* code legibility, which is why it is performed automatically in between software development and execution, at code distribution/deployment time. Due to its transformative nature code minification challenges the conventional notions of programming style, hence the interest of exploring its impact on the ability to automatically recognize code authors.

To investigate experimentally RQ 3 we still consider Python (for consistency and comparability with RQ 2) and use the Python Minifier library (*Flook, 2024*) as code minifier. Python Minifier is a popular source code minifier for Python, capable of applying transformations as varied as: renaming identifiers, removing white spaces, combining import statements, removing non-executable literals (*e.g.*, comments, docstrings), *etc.* For a comprehensive description of the modifications Python Minifier can apply please refer to the tool documentation at https://python-minifier.com/.

As an example, Listing 3 displays the result of minifying the code snippet of Listing 1 with Python Minifier.

## Author classifier

To answer the stated research questions we need a baseline author classifier. We build one based on code2vec (*Alon et al., 2019*), which transforms source code into vectors, ensuring the preservation of both its semantic and stylistic elements (due to the fact we feed CSTs, rather than ASTs, to it as already discussed).

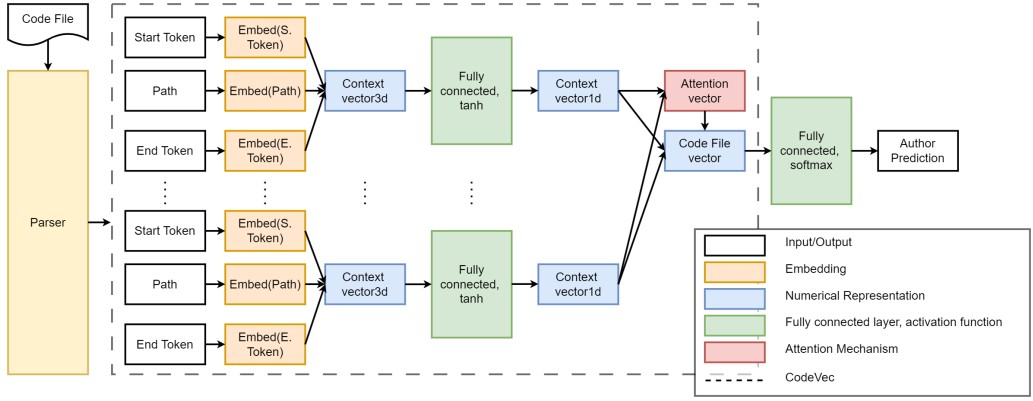

**Figure 3**  **Code2vec-based author classifier.**

The model functions independently of specific programming languages, allowing for a broad examination of code across different languages, drawing parallels to the language-independent nature of word2vec (*Mikolov et al., 2013*).

Once source code is transformed into vectors by code2vec, various tasks can be performed on it using apt machine learning techniques. In our case the task is determining the original author of a specific source code file. Such a task can be achieved with high accuracy, as substantiated by previous studies (*Kovalenko et al., 2020*). Note that prediction accuracy is an independent variable of this study: we are not trying to improve the state of the art in author predictability, but only to evaluate how it is impacted by code formatting and minification. Nonetheless, by building upon code2vec, we achieve a classification accuracy which is well within the state-of-the-art of code stylometry (cf. 'Results' for details).

Our author classifier takes as input a source code file, parses it to obtain a bag of path contexts, transforms them into vectors that represent the original file, and eventually produces as output an author prediction. The architecture of our classifier, shown in Fig. 3, consists of five parts:

- **Embedding layers**: each element of the input bag of path context triples undergoes individual embedding.
- **Concatenation**: the three elements of each path context triples are concatenated to obtain a unique representation, called *context vector3d*.
- **Fully connected layer**: each context vector3d is transformed to a one-dimensional representation, called *context vector1d*.
- **Attention layer**: all context vector1d elements of a code file are consolidated into a singular vector, called *code file vector*.
- **Fully connected layer**: this layer performs the final author prediction, based on the code file vector and using a softmax activation function.

## METHODOLOGY

Figure 4 depicts the experimental methodology followed to answer RQ 2 and RQ 3. The general idea is to, first, apply the author classifier of Fig. 3 (re-training it from scratch each

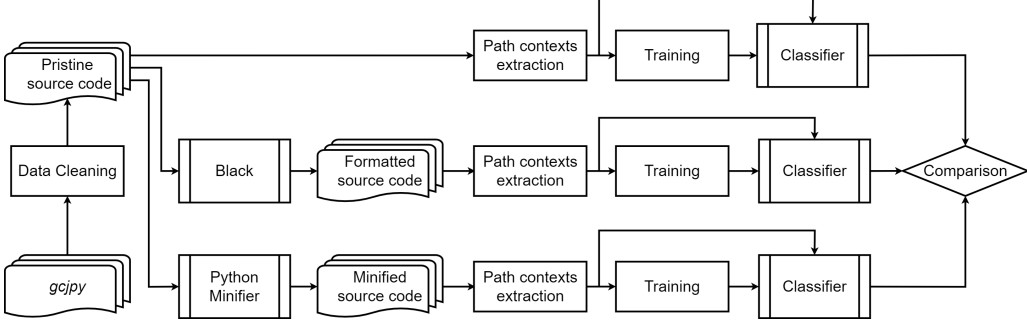

**Figure 4** Experimental methodology.

time) to: (1) a dataset of Python source code files equipped with an authorship ground truth, (2) the same dataset automatically formatted with Black, (3) the same dataset minified with Python Minifier. Second, compare accuracy results in the three cases. In the remainder of this section we detail each phase of the experiment implementation.

## Dataset

As initial dataset we started from the Google Code Jam dataset (*Google, 2023*) due to its prominence in code stylometry and its adoption in preceding studies (*Islam et al., 2015*; *Tereszkowski-Kaminski et al., 2022*; *Alsulami et al., 2017*; *Kurtukova, Romanov & Shelupanov, 2020*; *Bogomolov et al., 2021*). Using a shared dataset eases comparing our results and evaluation with previous works.

Google Code Jam is a coding competition where developers provide solutions to common problems in various programming languages. In the specifications of the competition, no instructions regarding style, guidelines or formatting are provided, thus each author is free to employ their preferred style. We focus on the Python subset of the Google Code Jam dataset, called *gcjpy* in the following. We obtained a copy of *gcjpy* from the replication package of *Bogomolov et al. (2021)* (discussed in 'Related Work') whom, in turn, scraped it from the Google Code Jam website.

*gcjpy* source code files are organized by authors; the organization hence implicitly provides a ground truth for the code →author mapping. The dataset contains code authored by 70 authors, 10 source code files per author, for a total of 700 Python files (cf. the *gcjpy* line in Table 1, which also reports average SLOC (source lines of code) and character lengths). Each file in the dataset contains a solution to a challenge presented, and solved, by all authors in the dataset.

The dataset is balanced both quantitatively (each author contributes 10 files) and semantically, with each author providing solutions to the same problems. The second aspect of balancing addresses the problem identified by *Islam et al. (2015)*, wherein authors could be classified based on the semantic content of their code rather than their unique styles. In ML-based code stylometry there is indeed a significant risk that the classifier might inadvertently learn semantic code attributes, overshadowing stylistic differences. By

**Table 1  Datasets used in the experiments.** For each dataset the following measures are reported: number of distinct authors, number of source code files (total), average SLOC (source lines of code) length, average character length.

| Dataset | Authors | Files | SLOC (avg.) | Characters (avg.) |
|---|---|---|---|---|
| gcjpy | 70 | 700 | 62 | 1609 |
| Pristine | 59 | 590 | 62 | 1561 |
| Formatted | 59 | 590 | 68 | 1655 |
| Minified | 59 | 590 | 25 | 979 |

analyzing files with uniform semantic content across authors, our classifier is forced to learn author stylistic differences, in order to improve its accuracy.

## Data cleaning

Code formatting and minification tools generally expect input source code to be syntactically correct; both Black and Python Minifier do. We hence applied a data cleaning step meant to verify that each file in the dataset is syntactically-correct Python code.

Out of a total of 700 initial files in *gcjpy*, 63 files (9%) by 11 distinct authors (15% of the initial 70 authors) contained at least one syntax error. Errors included misplaced blanks and keywords, as well as misclassified C source code files shipped with a .py extension. Differently from *Li et al. (2022)* we did not attempt to correct even the most obvious syntax errors, in order to retain the dataset's inherent integrity. This choice is consistent with what the majority of the literature on code stylometry do.

We excluded all non-parsable files from the dataset. In order to preserve dataset balancing we also excluded any author with less than 10 valid source code files remaining. This decision enables consistent analysis across all authors avoiding the need of synthetic data generation (*Batista, Prati & Monard, 2004*) to reestablish balance.

The dataset obtained after data cleaning, called *Pristine*, hence consists of 590 source code files, authored by 59 authors, 10 files each; cf. the Pristine line in Table 1.

## Code formatting

We applied the Black code formatter to each file in the Pristine dataset, obtaining the *Formatted* dataset. No errors were reported: Black could format all files in the Pristine dataset.

As observable in Table 1 (line Formatted), code formatting led to an increase in the average file length, both in terms of SLOCs (+10%) and characters (+6%). This change can be ascribed to Black's coding conventions which, with respect to original author styles in *gcjpy*, introduce additional white spaces, tabs, and newline characters.

## Code minification

We applied Python Minifier to each file in the Pristine dataset, obtaining the *Minified* dataset. Once again no error were reported.

As expected and shown in Table 1 (line Minified), minification significantly reduced the size of files in the dataset: −60% SLOC, −37% characters. This highlights quantitatively

the transformative nature of the minification process, already observed qualitatively by comparing Listing 1 with Listing 3.

## Path contexts extraction

To extract, prepare, and represent path contexts (see 'Code representation'), we implemented a language-agnostic pipeline based on the Tree-sitter parser generator tool and incremental parsing library (*Brunsfeld, 2024*). Tree-sitter comes with predefined grammars for many languages, including Python, and provides a unified API to navigate parse results. Tree-sitter is also well suited to manipulate *concrete* syntax trees, which we need, because it maintains a precise mapping from parse tree nodes to the original tokens and positions in the input file.

For each source code file in each dataset we proceed as follows. First, we use the Python Tree-sitter parser to parse the file and generate its CST. Then we traverse the CST to list all terminal nodes. During the traversal we filter out comments in order to avoid recognizing authors based on peculiar natural language content rather than programming style aspects.

We then generate all combinations without repetition of ordered terminal node pairs $\langle n_s, n_t \rangle$. Each pair consists of a starting terminal $n_s$ and an ending terminal $n_t$. Pairs are ordered in the sense that we only generate pairs where $n_s$ occurs earlier than $n_t$ in byte order in the input file.

For each pair a CST path $p$ is generated by navigating the CST upwards from $n_s$ until the least common ancestor (LCA) node of $n_s$ and $n_t$ is reached and then downwards from the LCA to $n_t$. From each obtained path $p$ we obtain a path context $\langle \phi(n_s), \text{sha256}(p), \phi(n_t) \rangle$, where $\phi$ is implemented by looking up the original input token associated to a node *via* Tree-sitter. The result is a path context as per Definition 3.

For practical purposes when extracting paths from a given input file we limit ourselves to paths with a maximum length of 7, following in the steps of the original code2vec implementation (*Alon et al., 2019*). The intuition behind this restriction is that code style is better captured by reoccurring local patterns (*e.g.*, if/then/else branches, loop bodies, *etc.*) than rarer global patterns, and that short paths captures local patterns best.

All path contexts extracted from an input file form its initial bag of path contexts as per Definition 4. To keep data size manageable during classifier training, we randomly sample a maximum of 200 path contexts from all contexts extracted from each file. On average, the number of path contexts extracted per file before sampling is 7,140 for the Pristine dataset, 6,778 for the Formatted dataset, and 7,413 for the Minified dataset. Once the path contexts have been extracted (and possibly sampled), we combine them to obtain the bag of path contexts for each file.

## Comparative analysis of AST *vs* CST

In order to address RQ 1, we conducted a parallel experiment replicating the methodology of *Bogomolov et al. (2021)*. Performing this replication with a controlled experiment on the same dataset is crucial for comparing the effectiveness of AST- *vs* CST-based code stylometry.

For this replication, we utilized the *Pristine* dataset derived from the *gcjpy*, as detailed in 'Data cleaning'. Note that the *gcjpy* version is the same as that used in *Bogomolov et*

*al. (2021)*. The architecture for this experiment, shown in Fig. 3, also mirrors that used by Bogomolov et al., using either the `ast` module of the Python standard library (for AST-based code stylometry) or Tree-sitter (for CST-based).

The AST-based experiment followed a similar procedural structure to our CST approach: the source code files from the *Pristine* dataset were parsed to generate their corresponding ASTs. Subsequently, the obtained ASTs were used to extract syntactic features which formed the basis of the input for the code2vec model. The model, trained on these features, was then employed to classify code authors.

## Model training

For each of the three datasets (pristine, formatted, minified) we train the author classifier of Fig. 3 from scratch and evaluate its accuracy. The model is trained on data for 59 authors, each having authored 10 files. We distribute the 10 files of each author as follows: six to the training set, two to the validation set, two to the testing set. This approach, with no overlap between the sets, guarantees the robustness and generalizability of our results.

Utilizing a grid search strategy, we optimize the model hyperparameters with Tune (*Liaw et al., 2018*), focusing on embedding dimension, batch size, number of training epochs, and dropout rate. Notably, higher dropout values exhibited superior performance. For the optimal configurations, refer to Table Table 2. For the optimization of model parameters during training, we use the ADAM optimizer (*Kingma & Ba, 2015*); to measure model efficacy and steer the optimization we use Cross Entropy Loss as primary metric, consistently with the multi-class classification nature of the problem.

Training and evaluation was conducted on a high-performance computing (HPC) cluster consisting of 10 processing nodes. Each node was equipped with an Intel Common KVM CPU with 4 cores operating at 2194.842 MHz, 44 GiB of RAM, and an Nvidia GeForce RTX 2080Ti 11 GiB graphics card. The average training duration for each dataset, using only a single node, was of 11 h, 34 min, and 14 s.

## RESULTS

We can now compare the accuracy results obtained by the author classifiers trained, respectively, on the three datasets: Pristine (using both ASTs and CSTs), Formatted, and Minified. Consistently with established literature, we use top-1 accuracy as primary evaluation metric (*Grandini, Bagli & Visani, 2020*).

## General results

On the Pristine dataset (*before* code formatting or minification, and using CSTs) the trained classifier achieves an average accuracy of 67.86%, which is comparable with the state-of-the-art for code2vec-based code stylometry on the *gcjpy* dataset. Table 3 provides a comparative overview of our results and related work; several observations are in order.

Our author classifier based on CST (bottom line in Table 3) performs slightly worse (−4.44%) than that of *Bogomolov et al. (2021)*. The difference is small enough to be imputable to differences in hyperparameter tuning during network design or training. But we also observe that, methodologically, we enforce a strict separation between training,

**Table 2  Best hyperparameter configuration for each dataset.**

| Dataset | Batch size | Dropout | Embed. Dim. | N. Epochs |
|---|---|---|---|---|
| Pristine source code (CST) | 8 | 0.7 | 64 | 300 |
| Formatted source code | 64 | 0.7 | 32 | 300 |
| Minified source code | 32 | 0.7 | 32 | 300 |

**Table 3  Comparison of authorship recognition results across approaches and related work.**

| Research | Approach | Accuracy |
|---|---|---|
| *Islam et al. (2015)* | random forest | 53.91% |
| *Alsulami et al. (2017)* | LSTM | 88.86% |
| *Bogomolov et al. (2021)* | code2vec | 72.30% |
| This work (pristine dataset, AST) | code2vec | 51.00% |
| This work (pristine dataset, CST) | code2vec | 67.86% |

test, and validation sets, whereas they use k-fold cross validation and grid search, which incurs a higher risk of overfitting than our approach. We confirm that code2vec-based author classifiers under perform (in mere accuracy terms, as they have other advantages) with respect to the LSTM-based one by *Alsulami et al. (2017)*; this is not problematic to pursue our research questions: what matters is using the same baseline classifiers on the three dataset variants.

## Impact of moving from AST to CST on authorship recognizability

By comparing the results of the AST-based and the CST-based classifiers on the Pristine dataset, we can answer experimentally RQ 1 (*What is the impact of using CSTs (instead of ASTs) on the ability to automatically identify authors of Python source code?*). The obtained respective accuracies can be observed on lines 4 and 5 of Table 3. It is indeed the case that in our experiments **moving from ASTs to CSTs leads to a significant increase in author recognizability: +16.86%**.

This finding aligns with the work of *Islam et al. (2015)*, confirming that layout features, which are not captured by ASTs, significantly contribute to the definition of individual programming style.

## Impact of code formatting on authorship recognizability

Moving on, we can now answer quantitatively RQ 2 (*What is the impact of code formatting on the ability to automatically identify authors of Python source code?*) by comparing the average accuracy of the classifier trained on the Pristine dataset and that of the classifier trained on Formatted dataset. Looking at the first two rows of Table 4 we can see that **the impact of automated code formatting** of Python code with Black **on author recognizability is significant**: the average accuracy after code formatting is 52.68% marking a decline of −**15.18%** with respect to the original code (67.86%).

The observed reduction suggests that automated code formatting partly "erases" source code features that are part of the stylistic signature of individual authors, making code more uniform across different authors, and hence code stylometry harder.

**Table 4** Accuracy of authorship recognition on different datasets: pristine (both CST and AST), formatted with Black, and minified with Python Minifier.

| Dataset | Accuracy (test set) |
| --- | --- |
| Pristine source code (AST) | 51.00% |
| Pristine source code (CST) | 67.86% |
| Formatted source code | 52.68% |
| Minified source code | 50.00% |

Nonetheless, **authors remain largely recognizable after code formatting**: an accuracy of $\approx 50\%$ across 59 authors is very significant—with the baseline of random attribution sitting at $1/59 \cdot 100 = 1.69\%$.

## Impact of code minification on authorship recognizability

What about code minification: would it make authors less recognizable than code formatting, and then nothing at all? We can now answer quantitatively RQ 3 (*What is the impact of code minification on the ability to automatically identify authors of Python source code?*) as well, by comparing the accuracy of the classifier trained on the Minification dataset and the two others: Pristine and Formatted; see Table 4, last line.

With respect to the Pristine dataset, once again **minification reduces author recognizability significantly: −17.86%**, from 67.86% down to 50.00%. We remind that code minification optimizes code size by stripping superfluous characters without altering its functionality, which includes deleting white spaces, comments, and renaming variables. Clearly those modifications impact on source code traits that embody a significant part of individual authors' coding styles, making them harder to recognize *via* code stylometry.

On the other hand, and perhaps surprisingly, the **decrease in author recognizability between code formatting and minification is minimal: −2.68%**, from 52.68% to 50.00% in our experiments. This is in contrast with the respective visual impacts of the two techniques when applied to real-world code: formatted code is generally perceived as being (1) not that different from before-formatting code and (2) *easier* to read; whereas minified code is considered much *harder* to read than before-minification code by most programmers. It appears that state-of-the-art code stylometry is largely immune to those differences and that most of the identifiable traits that could be suppressed automatically by these two techniques were *already* suppressed by Black, without needing to resort to Python Minifier for a significant further reduction in author recognizability. Furthermore, minification entails some form of code "uniformization" (*e.g.*, removing non-significant spaces) as one of its preliminary steps, which can be seen as a form of code formatting, even though not one geared towards making code easier on human eyes. In our experiments CST-based code stylometry does not appear to be particularly sensible to the differences in formatting style that exist between (Black) code formatting and (Python Minifier) minification.

Due to the small difference in impact between the two techniques, as before **authors remain largely recognizable after code minification** too, with a significant (for a set of 59 authors) success rate of 50% recognized authors. Once again, authors of software that must

be distributed in source code form cannot rely on code minification to be unrecognizable *via* stylometry techniques.

## DISCUSSION

### Relevance of the findings

The answers provided for the stated research questions inform the discussion around what contributes to author recognizability for software distributed in source code form.

In our controlled experiment, transitioning from AST to CST significantly improved style recognition. This aligns with Caliskan's conceptualization of programming style, which is broadly recognized as integrating *layout* elements along with *lexical* components, present in the CST but not in the AST. Our results suggest that, by capturing a more comprehensive spectrum of stylistic elements, CSTs can improve the accuracy of author recognition techniques, possibly beyond the state of the art when used in combination with already well performing methodologies, like those of *Alsulami et al. (2017)*.

Note that we replicated the experiment by *Bogomolov et al. (2021)* applying data cleaning, which was needed to ensure successful parsing. This reduced the dataset size, which may explain the lower absolute accuracy we obtained in our AST-based experiment with respect to theirs. As we compare, with a controlled experiment on the same dataset and in the same conditions, the use of CSTs and ASTs for code stylometry, we consider the accuracy drop from ASTs to CSTs both relevant and interesting in itself.

Both code formatting and minification reduce author recognizability by a significant margin, although not enough to make authors safe from de-anonymization. Other techniques are needed for that, such as full-blown obfuscation (*Kurtukova, Romanov & Shelupanov, 2020*); although in that case it would be harder to justify insisting on source code distribution *versus* the distribution of bytecode or binary code, which suppress most code style traits.

It is also worth noting that the accuracy of CST-based author recognition *after* code formatting or minification—respectively 52.68% and 50.00%, see Table 4—is very close ($\pm1.7$%) to the accuracy of *AST-based* author recognition on the pristine dataset at 51.00%. This suggests that formatting and minification *neutralize* the benefits obtained from including concrete syntax features in authorship attribution, without impacting the more intrinsic aspects of author style captured by abstract syntax features.

In terms of absolute accuracy results, even though improving the state of the art on that front was not a goal of this study, we are in the ballpark of previous approaches based on similar architectures (code2vec), which we consider satisfactory. At the same time, this is the first study that isolates by construction the effects of code formatting and minification from other effects that can become confounding variables for the stated conclusion. For example, whereas *Kurtukova, Romanov & Shelupanov (2020)* also observed a decrease in the recognizability of authors contributing to projects that adopt strict coding guidelines, the result might come from other project-specific factors. In our case the application of formatting and minification to the same initial code base, together with the training and validation of the same classifier, provide guarantees that the measured difference is only imputable to source code changes and not other incidental factors.

## Threats to validity

In the remainder of this section we review the adopted experimental methodology, in order to assess the threats to the validity of stated conclusions.

*Internal validity.* Our dataset of choice is the Python subset of the Google Code Jam dataset (*Google, 2023*), a popular choice for benchmarking authorship attribution solutions (*Islam et al., 2015*; *Tereszkowski-Kaminski et al., 2022*; *Alsulami et al., 2017*; *Kurtukova, Romanov & Shelupanov, 2020*; *Bogomolov et al., 2021*). This choice is motivated in the literature with two main arguments: result comparison and mitigating biases coming from the domain/project of origin of input source code file. Result comparison is a common need for studies that aim to improve code stylometry accuracy: they need to compare new solutions with previous ones on a common benchmark; we do not share this need due to different study goals. Domain bias is the risk of letting the ML model inadvertently learn that two code snippets code belong to, respectively, the Linux kernel and Nginx, and mistaking that from having learned they belong to the code style of the respective authors who only contribute to one of the two projects.

Regarding the foundation on which the model learns, employing the Google Code Jam dataset ensures that the models do not inadvertently learn to associate code content with specific authors. This means that the model is designed not to merely recognize what an author typically writes, such as code related to web development, data manipulation functions, or numerical calculations. Instead, the model is cultivated to discern and learn the distinct programming styles of each author genuinely. This unique characteristic, previously highlighted by Caliskan, is achieved due to the nature of the dataset, which requires each author to respond to the same set of challenges by writing code files, all while using the same programming language. By using *gcjpy* we avoid this risk of this bias, as all retained authors have contributed the same set of 10 snippets, each proposing a different solution for the same problem.

In the extraction pipeline, we implemented the following steps: excluding files that include syntax errors, under-sampling the dataset to maintain balance, extracting path contexts, and finally applying formatting or minification. Our cleaning process is highly conservative, ensuring that all steps are aligned to preserve a balanced dataset, honoring a strict training/validation/testing set separation.

Path extraction introduces some challenges. One concern emerges from the random sampling of 200 path contexts to maintain a fixed maximum size of path bags, as recommended by code2vec. This constraint results in some file representations that incorporate only a small fraction, between 0.0269% and 0.0295%, of all file paths. The sampling being random, we have repeated our experiments 10 times, always obtaining results comparable with those reported in the article. We are hence confident the context sampling step does not introduce appreciable biases.

Similarly, the maximum limit on path lengths, also inherited from code2vec, might induce biases in our results: longer paths might make authors more recognizable. We have not performed a parameter sweep on path length to quantify this phenomenon. However,

as we use the same classifier architecture for all datasets, we do not expect it to impact the main experimental results of the article.

The choice of tools for formatting (Black) and minification (Python Minifier) is debatable like any tool choice. We are however confident to have chosen the tools that were considered state of the art for the respective tasks in the Python community at the time, as corroborated by our personal experience as Python developers, technical exchanges with professional Python developers, as well as Web searches for the respective keywords. We acknowledge that the best choice of tools can evolve over time. Furthermore, we make available a complete replication experiment for the results presented in this article, which would allow to reproduce our experiment in the future, possibly with different tools.

*External validity.* The stated research questions are about author recognizability for Python source code, consistently with the use of the *gcjpy* dataset. We do not claim more generality than that with respect to other programming languages. Differently from most related works, however, the experimental methodology is inherently language-agnostic. In particular the path context extraction pipeline available from the replication package can be applied to any programming language for which a Tree-sitter grammar exists (more than 100 parsers at the time of writing; Cf. https://tree-sitter.github.io/tree-sitter#parsers, accessed on 2023-10-25). The remainder of the experiments from there on (training, evaluation, comparison) would be repeatable without changes.

Previous works (*Bogomolov et al., 2021*; *Kovalenko et al., 2020*) hint at the fact that author recognizability is largely unaffected by programming language differences. Experimental evaluations of this fact are still lacking in the literature, though. We expect source-to-source code transformations like those we evaluate to be even less affected by programming language differences, given the language remains the same across transformations.

Our findings are inherently rooted in a controlled setting, specifically a programming competition where algorithmic problem solving is predominant in source code files. It would be imprudent (although quite common in the literature!) to speculate without empirical validation that our results seamlessly transfer across diverse coding domains. Some surveyed related works (*Bogomolov et al., 2021*; *Kovalenko et al., 2020*) provide a glimpse into analogous studies conducted within real-world development contexts, including multi-author Web development projects and applications. To establish the uniformity of results across different development domains it is imperative to replay our methodology in those scenarios. Yet, securing a balanced dataset in non-controlled coding situations remains a significant challenge.

## CONCLUSION

In the context of code stylometry techniques, used to automatically identify the author of a given source code snippet, we have characterized quantitatively the impact of popular software engineering practices like automated code formatting and minification on author recognizability. Experiments were conducted on the Python subset of the Google Code Jam dataset (59 authors), using Black as code formatter, Python Minifier as minifier, and a machine learning classifier based on code2vec.

The applied experimental methodology—namely: start from a clean dataset, transform it to obtain corresponding formatted and minified datasets, and then separately train author classifiers on the three datasets—provides strong guarantees of having measured the impact of formatting/minification, rather than conflating them with confounding factors. As a novelty with respect to the state of the art we fed code2vec with concrete syntax trees (CSTs), rather than ASTs, verifying how that increases classification accuracy in a controlled experiment.

Experimental results show that: (1) moving from AST to CST enhances by 17% author recognizability (51.00% → 68%), (2) code formatting reduces it by 15% (68% → 53%), (3) code minification further reduces it by 3% (68% → 50%), (4) none of the two techniques are enough to guarantee author non-recognizability. For that, more invasive techniques should be explored, such as obfuscation or compilation, giving up almost completely on code readability.

*Future work.* Comparing the effect of code formatting and minification on the recognizability of developers writing in different programming languages remains an open lead. Whereas empirical studies on code stylometry have been conducted on different programming languages, benchmark-style experiments (in terms of code size, problem solved, settings) are still missing, partly due to the difficulty of obtaining relevant balanced datasets.

A second area of extension of this work lies in enlarging the set of source-to-source code transformation considered. Obfuscation is here an obvious low-hanging fruit to evaluate next (in a clean-slate setting w.r.t. the original dataset, differently from related work on the topic *Kurtukova, Romanov & Shelupanov, 2020*), but other interesting possibilities exist, like automated code refactoring as presently available in modern IDEs.

### Funding
The authors received no funding for this work.

### Competing Interests
The authors declare there are no competing interests.

### Author Contributions
- Stefano Balla conceived and designed the experiments, performed the experiments, analyzed the data, performed the computation work, prepared figures and/or tables, and approved the final draft.
- Maurizio Gabbrielli conceived and designed the experiments, performed the experiments, authored or reviewed drafts of the article, and approved the final draft.
- Stefano Zacchiroli conceived and designed the experiments, performed the experiments, authored or reviewed drafts of the article, and approved the final draft.

## Data Availability

The replication package is available at Zenodo: BALLA, S., GABBRIELLI, M., & ZACCHIROLI, S. (2024). Code Stylometry vs Formatting and Minification - Replication Package. Zenodo. https://doi.org/10.5281/zenodo.10528246.

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
