# Peer review of "Code stylometry vs formatting and minification"

_PeerJ Computer Science, doi:10.7717/peerj-cs.2142_

## Round 0.1 · original submission · Major Revisions

The reviewers and I agree that this is an interesting and well-written manuscript. However, there are several suggestions for improving the paper that you need to take into account. Most importantly:

- Clearer and deeper motivation of research questions
- Clearer and explicit assumptions
- Discussion of the conclusions in contrast to assumptions
- Better description of study design choices

**Language Note:** The review process has identified that the English language must be improved. PeerJ can provide language editing services - please contact us at [email protected] for pricing (be sure to provide your manuscript number and title). Alternatively, you should make your own arrangements to improve the language quality and provide details in your response letter. – PeerJ Staff

Reviewer 1 ·

Basic reporting

There are some minor grammatical issues in various places in the paper. I would suggest a thorough proofread before final publication to improve readability.

Experimental design

no comment

Validity of the findings

While I found this paper informative and useful, I find the headline conclusions to be incompletely stated. The improvement of CSTs of ASTs is due to the inclusion of layout features. Your results for the two tested defenses bring the CST accuracy back down to the level of AST accuracy - effectively, these (trivial) defenses, while failing to anonymize the code, do remove the benefit gained from augmenting the AST-based features with layout features. In fact, this is precisely one of the reasons that AST-based features have become one of the standard techniques used - they are resilient against the easiest to perform obfuscating techniques (which you experiment on in this work). While you do mention this in your motivation for using CSTs, this is somewhat buried in comparison to the conclusions you call out throughout the paper. It is important to highlight not only the underlying assumption, but that your results clearly align with the assumption. Effectively, your work demonstrated three things, with only the first 2 clearly called out:
1) Layout features combined with AST-based features enhance attribution success
2) Formatting and minification do not successfully anonymize code
3) Formatting and minification counter the benefit gained by adding layout features, but do not hinder the use of AST-based features.
If you highlight this third observation, this paper will be clearer and will reduce the risk of being misleading.

Cite this review as
Anonymous Reviewer (2024) Peer Review #1 of "Code stylometry vs formatting and minification (v0.1)". PeerJ Computer Science

Reviewer 2 ·

Basic reporting

- Relevance of the problem is provided however a more clear goal (e..g, recognising the author of a code snippet with and without code formatting and minification see their impact).

- Introduction describes contributions that are not clear. Can you please list the contributions? Is the contribution a CST-based approach for code stylometry?

- Research questions are not well motivated.

- Replication package is provided.

- An example can be given in the start to help the readers or at least redirect to AST based and CST based example of Figure 1.

- Presentation concerns: Overall the quality of figures is good.
- Line 98: footnotes after punctuations.
- Line 302: highlight "gcjpy".

- Literature well referenced & relevant
Relevant sources are required to support the statement at multiple places, for example, line 48, 84 etc.

Experimental design

- Journal Scope: within scope.
- Research question relevant: the questions are relevant but not well motivated. Previous work has investigated similar but why the authors are interesting in replicating previous work, or their pipeline is not made clear.
- Conclusions are well stated, linked to original research question.
- Relevant sources are required to support the statement at multiple places, for example, line 48, 84 etc.
- Some study design choices need to be better justified.

Actionable remarks for overall papers.
* * *
Line 30: "Doing so is referred to as, interchangeably in the literature, authorship
attribution or code stylometry" -> cite the relevant sources where these terms are referred to as.

Line 38: "retrieving authorship information that went missing to productivity enhancements" -> Can we have the citiation next to the use case so it is easy to track and verify the sources.

Line 48: "Impact of various kinds of source code manipulation on authorship
49 attribution is, however, fairly limited." -> Can you please cite a few of those limited studies so that it is known what areas has been focused?

Line 50: "the impact of common software development practices on the accuracy of
51 code stylometry remains largely unexplored." -> What common development practices? e.g., would writing code comments be a practice?

Line 62-64: Would it be possible to have an example of AST and CST example to show the source-to-source transformation? or refer to figure 1.

- Research questions are not motivated well. For RQ1: Although it is briedfly mentioned about the working of CSTs vs ASTs, why are we considering CSTs and ASTs for authorship attribution? When stylishtic change do not impact ASTs, why are we comparing them against? it is unclear.
Please describe the motivation for other RQs as well. For example, is this the questions are whether with or without code formatting, identification of python source code is done easier?

Line 69: "While performing code stylometry on CST is a technical need of this work" -> it is not a strong motivation for the RQ.

Line 84: :"The two chosen tools, Black and Python Minifier, are popular and state-of-the-art tools for the respective tasks." -> Provide relevant sources to support the statement

Line 90: "neither is enough to make developers non-recognizable via code stylometry." -> is the goal to keep developers anonymous? or make developers anonymous? Unclear.

Line 122: "These features are crucial for quantifying the impact of code formatting and minification." -> Why are they crucial?

Line 149: "quantifying the effect of specific practices on author recognizable" ->
Why not to specify it in the introduction as well that there are various practices shown in previous work that impacts the author attribution, which means certain practices make it easy or difficult to detect the authors. We also aim to test the impact of various similar practices on authorship attribution.

In related work section, no related work to CSTs is given, is there no area in software engineering that uses it?

Line 232: There are other linters as well for Python; why is Black chosen and not others?

Listing 3: Does Minifier replaces only identifiers name or also the functions name as well?

Line 353: I understand the reason to filter comments for minification but why for formatting. When the aim is to see the impact of code formatting which includes all lexcial and layout components, then why comments are filteres as they also includes the mentioned component?

Line 363: selecting the maximum length of 7 is not well motivated. Can you please show an example of rarer global patterns? why the preference to capture local pattern over global pattern is given?

Line 367: how many total path contexts are there?

Line 389: what are embedding dimensions, batch size, number of training epochs, and dropout rate?

Line 408: "Our main author classifier (bottom line in the table)" -> Which table? table 2?

Line 409" "The difference is small enough to be imputable to differences in hyperparameter
410 tuning during network design or training." -> why the difference of 4% is considered small? While in other later cases, the performance improvement of 16% is claimed to be significiant?

Line 436: "Authors wishing not to be identified should not rely on code formatting alone as a defense mechanism against code stylometry." -> can you cite the relevant sources that shows that authors use it as a defense mechanism? how often is it used?

"decrease in author recognizability between code formatting and minification is minimal: -2.68%, from 52.68% to 50.00% in our experiments." -> On one side, the results show that layout has a significant impact on code stylometry. In contrast, on the other side the difference is minimal between formatting anf minification (which exclude layout features). Can authors shed more light on it?

There can be various coding style guidelines that developers can follow, for example PEP257, Numpy etc. Which of those guidelines are followed by developers in this dataset is neither considered nor discussed. Similarly there can be various factors such as the size of code base, project community standard. Developers can follow different styles when they are doing personal projects or the open source projects.
It is also not described what coding style guidelines Black follows. Many times, the coding guidelines from various sources can be contradictory with each other or with the tool. In such a case, Black might not be useful.

Line 512: How do authors never the 10 snippets by each developers are different solutions for the same problem? Clarify.

Validity of the findings

A replication package is provided.

Cite this review as
Anonymous Reviewer (2024) Peer Review #2 of "Code stylometry vs formatting and minification (v0.1)". PeerJ Computer Science

---

## Round 0.2 · accepted · Accept

As the reviewers are satisfied with the changes and I do not see any remaining problems, I am happy to recommend the manuscript for publication.

Reviewer 1 ·

Basic reporting

no comment

Experimental design

no comment

Validity of the findings

no comment

Additional comments

I find this version of the manuscript significantly improved over the previous submission and believe that the comments on the previous submission have been thoroughly and thoughtfully addressed. I have no remaining concerns and believe that this work will benefit the state of knowledge on this important topic and help get the community closer to understanding the factors involved with both attribution and anonymization.

Cite this review as
Anonymous Reviewer (2024) Peer Review #1 of "Code stylometry vs formatting and minification (v0.2)". PeerJ Computer Science

Reviewer 2 ·

Basic reporting

no comment

Experimental design

- Journal Scope: within scope.
- Research question relevant: the questions are relevant and motivated.
- Claims are supported with references.

Validity of the findings

no comment

Cite this review as
Anonymous Reviewer (2024) Peer Review #2 of "Code stylometry vs formatting and minification (v0.2)". PeerJ Computer Science